

# High-resolution global atmospheric moisture connections from evaporation to precipitation

Obbe A. Tuinenburg[1], Jolanda J.E. Theeuwen[1,2], Arie Staal[1,3]

[1] Copernicus Institute of Sustainable Development, Utrecht University, Utrecht, 3508 TC, the Netherlands
[2] Wetsus, European Centre of Excellence for Sustainable Water Technology, Leeuwarden, 8911 MA, the Netherlands
[3] Stockholm Resilience Centre, Stockholm University, Stockholm, SE-10691, Sweden

*Correspondence to*: Obbe A. Tuinenburg (o.a.tuinenburg@uu.nl)

**Abstract.** A key Earth system process is the circulation of evaporated moisture through the atmosphere. Spatial connections between evaporation and precipitation affect the global and regional climates by redistributing water and latent heat. Through this atmospheric moisture recycling, land-cover changes influence regional precipitation patterns, with potentially far-reaching effects on human livelihoods and biome distributions across the globe. However, a globally complete dataset of atmospheric moisture flows from evaporation to precipitation has been lacking so far. Here we present a dataset of global atmospheric moisture recycling on both 0.5° and 1.0° spatial resolution. We simulated the moisture flows between each pair of cells across all land and oceans for 2008−2017 and present their monthly climatological means. We applied the Lagrangian moisture tracking model UTrack, which is forced with ERA5 reanalysis data on 25 atmospheric layers and hourly wind speeds and directions. Due to the global coverage of the simulations, a complete picture of both the upwind source areas of precipitation and downwind target areas of evaporation can be obtained. We show a number of statistics of global atmospheric moisture flows: land recycling, basin recycling, mean latitudinal and longitudinal flows, absolute latitudinal and longitudinal flows, and basin recycling for the 26 largest river basins. We find that, on average, 70% of global land evaporation rains down over land, varying between 62% and 74% across the year; 51% of global land precipitation has evaporated from land, varying between 36% and 57% across the year. Highest basin recycling occurs in the Amazon and Congo basins, with evaporation and precipitation recycling of 63% and 36% for the Amazon basin and 60% and 47% for the Congo basin. These statistics are examples of the potential usage of the dataset, which allows users to identify and quantify the moisture flows from and to any area on Earth, from local to global scales. The dataset is available at https://doi.pangaea.de/10.1594/PANGAEA.912710 (Tuinenburg et al., 2020).

## 1 Introduction

Atmospheric moisture flows from evaporation to precipitation are a fundamental component of the hydrological cycle on Earth (Brubaker et al., 1993; Savenije, 1996; Van der Ent et al., 2010). It is estimated that 36% of global rainfall over land evaporated from land (Van der Ent et al., 2014) and that most evaporation is transported for several hundreds to thousands km before





precipitating (Dirmeyer and Brubaker, 2007; Van der Ent and Savenije, 2011), although the regional orography can influence this significantly (Tuinenburg et al., 2012). This means that land-cover changes that affect evaporation have regional consequences. Indeed, it is increasingly recognized that anthropogenic land-use changes affect atmospheric moisture flows and our dependencies on them (Keys et al., 2016, 2019; Wang-Erlandsson et al., 2018; Staal et al., 2020). However, the remote effects of evaporation changes on precipitation patterns remain surprisingly poorly understood, partly due to limited systematic
documentation of atmospheric connections.

Often, studies on the atmospheric component of Earth's hydrological cycle use moisture tracking models based on atmospheric reanalysis data to simulate the flows of moisture (Dirmeyer et al., 2009; Gimeno et al., 2010; Van der Ent and Savenije, 2011; Tuinenburg et al., 2012). The recent development of the ERA5 reanalysis data (Copernicus Climate Change Service, C3S) allows for unprecedented high-resolution estimates of moisture flows between any pair of areas across the globe. However,
the required data input for high-quality moisture flow simulations with ERA5 can be high (Tuinenburg and Staal, 2020), which may preclude wide application of these simulations. A readily usable and comprehensive global-scale dataset of pairwise links of atmospheric moisture connections could satisfy much of the demand, but it has been lacking until now. Here we present such a global-scale dataset of moisture flows between all pairs of land and ocean surface at 0.5° and at 1.0° spatial resolution. Its resolution of 0.5° means that the dataset divides the Earth into 259,200 grid cells with moisture flows between each pair of
them, while the one at 1.0° resolution has 64,800 cells. Together, the flows from any evaporation source cell to all its target (precipitation) cells represent the 'footprint' of the source cell. This dataset consists of monthly multi-annual mean footprints for 2008−2017.

The dataset was generated with 'UTrack-atmospheric-moisture', hereafter 'UTrack', which is a Lagrangian (trajectory-based) moisture tracking model developed by Tuinenburg and Staal (2020). In the simulations, large amounts of moisture particles
were released at random locations and heights within each cell, after which their locations were tracked based on the reanalysis data for wind speed and direction. Thus, whereas the presented output is grid-based, the simulations behind them are not. The UTrack model behind this dataset resulted from an elaborate sensitivity analysis to test the effects of a range of assumptions and uncertainties on the accuracy of moisture tracking simulations and hydrologically relevant statistics (Tuinenburg and Staal, 2020).

Our dataset is the first to present global moisture flows across the globe, including both land and ocean. Although the simulations were carried out using 'forward tracking' from evaporation to precipitation, it implicitly includes the backward trajectories of moisture flows. Because our dataset offers a complete picture of moisture flows across the globe including the oceans, it can be used to construct both forward footprints or 'evaporationsheds', the downwind areas of precipitation receiving evaporation from regions of interest (Van der Ent and Savenije, 2013), as well as backward footprints or 'precipitationsheds',
the upwind areas of evaporation contributing to precipitation in regions of interest (Keys et al., 2012). Furthermore, it is the first to use the new ERA5 atmospheric reanalysis data and the most detailed to date.



This paper is structured as follows. In Section 2 we outline the methods behind the data, where we summarize the UTrack model in Section 2.1, detail the global simulations and the format of their output in Section 2.2, and explain our data validation procedure in Section 2.3. In Section 3 we present a number of results regarding the global synthesis of atmospheric moisture

flows. First, in Section 3.1 we show a number of metrics obtained from the data: evaporation moisture recycling (land and basin recycling) and average travelled distance (longitudinal and latitudinal direction). Next, in Section 3.2 we present validating tests of the dataset: sum of precipitation originated from any grid cell, which should add up to one, and comparison of total precipitation of UTrack and ERA5. We briefly relate this dataset to existing work and discuss its potential applicability in Section 4. Section 5 presents the conclusions and Section 6 gives the instructions to download the data and the scripts to

process them.

## 2 Methods

### 2.1 Atmospheric moisture tracking model

The dataset is generated using the Lagrangian atmospheric moisture tracking model UTrack by Tuinenburg and Staal (2020) (Figure 1). The model tracks parcels of moisture through the atmosphere from their locations of evaporation to those of

precipitation, based on ERA5 atmospheric reanalysis data. This means that the model post-processes these atmospheric data. The moisture tracking consists of three steps: 1) The release of moisture evaporated from the land surface into atmospheric moisture parcels, 2) the calculation of trajectories through the atmosphere for each parcel, and 3) the allocation of moisture present in the parcels to precipitation events at the location of the parcel.

In the first step, ERA5 total evaporation is determined and divided into atmospheric moisture parcels. For each mm of evaporation, 100 parcels are released 50 hPa above the surface height at random spatial locations within each $0.25° \times 0.25°$ cell across the globe. To make sure our estimates are representative of all evaporation and do not exclude locations or moments with small amounts of evaporation, at least one moisture parcel is released per hour for each location, even if there is less than 0.01 mm of evaporation.


In the second step, the parcels are tracked forward in time through three-dimensional space, where the location and moisture content of each parcel are updated at every time step of 0.1 h. This forward trajectory calculation is based on interpolated three-dimensional ERA5 wind speed and wind direction data, which consist of 25 layers in the atmospheric column. This step is partly randomized as follows. The main uncertainty of atmospheric moisture tracking is the redistribution of moisture in the

vertical direction (Tuinenburg and Staal, 2020), for example due to parameterized processes including convective up- and downdrafts, re-evaporation, and microphysics. To account for the uncertainty in the vertical redistribution, we employ a probabilistic scheme in which moisture parcels are randomly distributed along the local vertical moisture profile (Tuinenburg





and Staal, 2020). The scheme is run at every time step, with such a probability that every parcel is redistributed along the local moisture profile on average once per 24 h.


In the third and final step in the moisture tracking model, a fraction of the moisture present in a moisture parcel may be allocated to rainfall events at the location of the parcel. For this, ERA5 hourly total precipitation ($P$) and total precipitable water ($TPW$) are used, which are interpolated to the simulation time step of 0.1 h. At every time step, the amount that rains out is equal to the amount of precipitation at that time step over the total precipitable water in the water column ($P / TPW$).


Precipitation $A$ (mm per time step) at location $x,y$ and time $t$ that has evaporated from any location of release in any cell is equal to:

$$A_{x,y,t} = P_{x,y,t} \frac{W_{parcel,t} E_{source,t}}{TPW_{x,y,t}},$$   eq. 1


where $P$ is rainfall in mm per time step, $W_{parcel}$ is the water in the tracked parcel in mm, $E_{source}$ is its fraction of water that evaporated from the source, and $TPW$ is the precipitable water in the atmospheric water column in mm. At each time step, the moisture content of the parcel is updated based on evapotranspiration $ET$ into the parcel and rainfall $P$ out of it:

$$W_{parcel,t} = W_{parcel,t-1} + \left(ET_{x,y,t} - P_{x,y,t}\right) \frac{W_{parcel,t-1}}{TPW_{x,y,t}}.$$   eq. 2

The fraction of water in the parcel that has evaporated from the source then becomes:

$$E_{source,t} = \frac{E_{source,t-1} W_{parcel,t-1} - A_{x,y,t}}{W_{parcel,t}}.$$   eq. 3


Thus, the amount of water that was tracked from the source location decreases with precipitation along its trajectory. Each parcel is tracked for 30 days or until 99% of its moisture content has rained out. The moisture flow $m_{ij}$ in mm month[-1] that links evapotranspiration in cell $i$ to rainfall in cell $j$ where $[x,y] \epsilon j$ becomes:

$$m_{ij} = \sum_{t=0}^{month} A_{j,t} \frac{ET_{i,t}}{W_{i,t}},$$   eq. 4


over the course of a given month. Here, $ET_{i,t}$ is the evapotranspiration in mm per time step; $W_{i,t}$ is the tracked amount of water from source cell $i$ at time step $t$.



Further information about the justifications of the model settings can be found in Tuinenburg and Staal (2020).

## 2.2 Global moisture flow simulations

For all ERA5 moisture evaporated between 2008 and 2017, we determined the downwind precipitation location using UTrack. These simulations were done globally for each 0.25° ERA5 grid cell, including the North and South poles. Although moisture was released from each 0.25° grid cell and the simulations were forced by data at 0.25° resolution, we aggregate this monthly

and provide output at 0.5° and 1.0° resolution. This degradation of the spatial resolution is due to the large data load of a global pairwise set at 0.25°. We also reduced the size of the data set by providing monthly means over 2008−2017 (weighted by the monthly evaporation in these years) rather than providing complete time series. This multi-annual monthly mean result of these simulations is the presented estimate of the downwind precipitation for the evaporation for each grid cell. This downwind precipitation is then normalized, so that the global downwind sum for each grid cell equals one.


In order to save storage space, the output is treated as follows. First, we take the natural logarithm of the downwind locations and we subsequently multiply this by -10. Then, these results are converted from their internal data type of floating point numbers with double precision to unsigned integers (range 0−255) and stored in a NetCDF4 file, with compression level 6. This procedure limits file sizes considerably, but it comes at the cost of some imprecision due to the conversion of doubles to

integers.

The procedure to retrieve the normalized downwind precipitation locations of evaporation at location $x,y$ ($P_{E\_xy}$) is:

$$P_{E\_xy\_database} = \exp\left(moisture\_flow[x,y,:,:] \times -0.1\right) \qquad \text{eq. 5}$$


Due to the imprecision of the storage in integers, the global sum of $P_{E\_xy\_database}$ typically is around 1.05. Therefore, this needs to be renormalized as follows:

$$P_{E\_xy} = \frac{P_{E\_xy\_database}}{\sum P_{E\_xy\_database}} \qquad \text{eq. 6}$$

## 2.3 Metrics and validation

In addition to the high-resolution moisture connections, this dataset includes metrics describing each monthly file in the database. The metrics that are included in the database are evaporation moisture recycling (land and basin recycling), precipitation moisture recycling (land and basin recycling), downwind distance between evaporation and precipitation location (longitudinal and latitudinal direction), global simulated precipitation, and sum of precipitation originated from one source

cell.





Land evaporation recycling (in short land recycling) is defined as the fraction of evaporation that precipitates over land areas, and basin evaporation recycling (in short basin recycling) is defined as the fraction of evaporation that precipitates in the same river basin it evaporated from. The land recycling is determined using the ERA5 land-sea mask. The river basin recycling is determined using basin data derived from the drainage direction map DDM30 (Döll and Lehner, 2002). Note that the river basin recycling rates depend on the size and shape of the river basin. Furthermore, all oceanic values and Antarctica are considered to be one hydrological unit and are thus the same 'river basin'.

The distance between evaporation and precipitation locations is determined for each grid cell from its evaporation footprint, determined by taking the mean over all downwind locations (weighted by moisture flow). For this mean distance, we present its latitudinal and longitudinal component. Because flows in opposite directions cancel out one another, we also present the mean absolute distances, which are more informative of the actual distances between evaporation and precipitation.

The above metrics are all related to the characteristics of the relation between evaporation and its downwind location, which result from the forward simulations that form the basis of the dataset. In addition to these, we employ the fact that the database contains the downwind information for evaporation from all locations globally. By multiplying the (normalized) downwind precipitation locations with the evaporation in the source location, we get the absolute amount of downwind precipitation. In this 'backward analysis', if the absolute downwind precipitation resulting from all global evaporation locations is summed, the total global precipitation should result. Moreover, we can determine the evaporation sources of precipitation in each individual grid cell by multiplying the normalized values in the database with the actual evaporation in each grid cell and only select the precipitation grid cell of interest from the downwind precipitation footprints. This will then result in the precipitation in the grid cell under consideration and its evaporation origins. Thus, for the evaporative sources of precipitation the precipitation moisture recycling (land and basin recycling) are determined.

Two tests are done on the dataset. First, to check for numerical consistency we determine whether the sum of precipitation originating from each source cell adds up to the evaporation from that cell. Note that the total sum can deviate from 1 due to the storage format, as described in Section 2.2. Second, for each grid cell, we multiply the (normalized) downwind precipitation location with the monthly evaporation for that grid cell, yielding the actual downwind precipitation from that grid cell. The sum of this actual downwind precipitation over all the grid cells releasing evaporation should add up to the global precipitation amount and pattern again. The global precipitation should be similar to ERA5 total precipitation as the dataset is forced with those data. Therefore, to check the internal consistency of the dataset, the global precipitation is calculated from the database and compared to ERA5 total precipitation.



## 3 Results

### 3.1 Global metrics

We find that, globally, 70% of all terrestrial evaporation precipitates over land (i.e. the global average land recycling ratio is 0.70). However, the land recycling ratio varies considerably across the globe (Figure 2). For example, in southern South America the land recycling ratio is low due to strong transport of moisture towards the ocean there. For inland regions, land recycling ratios are close to 1. Furthermore, the land recycling ratio varies throughout the year. The largest percentage of terrestrial evaporation recycling occurs in June and July (74%) and the smallest in January and December (62%) (Figure 3). The Northern Hemisphere contains more land surface and therefore, there is more convection during summer months in the Northern Hemisphere, which has an important contribution to the evaporation recycling ratio (Figure 2).

Only 17% of moisture evaporated from the ocean rains out over land. In general, the land recycling ratio for the oceans (the fraction of evaporation from an ocean location that rains out over land) does not exceed 0.2. Exceptions include the high land recycling around the equator as a result of the trade winds that transport oceanic moisture towards land. Also, both in the north-eastern and south-eastern Pacific Ocean and the northern Atlantic Ocean, land recycling exceeds 0.2 due to the westerlies (Figure 2A). Throughout the year there is little variation in the relative amount of moisture that evaporates from the ocean and rains out over land. There is a small deviation from the annual mean in the months April (16%), May (15%) and June (16%). All other months show no deviation from the annual mean (17%).

With the implicit backward analysis we find that 51% of global precipitation over land and 8% of precipitation over the ocean has evaporated from land. Figure 4 shows the seasonal variation of the ratio of precipitation with moisture evaporated from land. In the Northern Hemisphere, there is a clear seasonal cycle as, here, during spring and summer (AMJ and JAS) the land recycling ratio is larger as a result of convection, both over oceanic and terrestrial regions. For the Southern Hemisphere the seasonal differences are smaller. However, over Australia, the land recycling ratio is also larger for spring and summer (OND and JFM). As the seasonal cycle is stronger for the Northern Hemisphere, the global precipitation over land and ocean that evaporated from land show maximum values for June (57% and 10%) and July (57% and 10%) (Figure 3). Minima occur during the Northern Hemisphere winter. The part of global precipitation over land that evaporated from land is lowest in January (36%) (Figure 3); the part of global precipitation over the ocean that evaporated from land is lowest in January, February and March (7%).

Out of the 26 largest river basins of the world (Table 1), evaporation recycling ratio in almost all cases exceeds precipitation recycling ratio (Figure 5). Only for the Murray and Mackenzie basins we find a larger proportion of recycled precipitation than recycled evaporation (12% vs. 11% for the Murray, and 34% vs. 33% for the Mackenzie basin). Highest evaporation recycling ratios occur in the Amazon and Congo basins, with respective ratios of 63% and 60%, which can be explained by their size and shape, high forest evapotranspiration, and strong convection over the tropical rainforests (Spracklen et al., 2012; Staal et



al., 2018). The Congo basin has a higher precipitation recycling ratio than the Amazon: in the Congo, 47% of precipitation has last evaporated in the same basin, against 36% for the Amazon.

Lowest recycling ratios are found for the low mid-latitudes, with values between 0 and 0.4 (except for mountainous regions). Due to the Hadley cells and the Ferrel cells there is subsidence in the low mid-latitudes. As the cool air sinks it warms adiabatically, which results in a decrease in relative humidity. Because of this, the air is dry and precipitation is less likely to
occur, which explains the lower recycling.

Figure 6 shows the annual mean of the travelled distance for evaporated moisture across the globe. For latitudinal transport there is an equatorward flow for the polar regions and a poleward flow around the equator. For the mid-latitudes an alternating pattern of northward and southward moisture transport is visible, with maxima up to 18° in both directions. The annual mean of travelled latitudinal distance for the backward analysis shows the same pattern (Figure 7). However, the magnitude of this
transport is smaller, as the maximum values go up to only 10°. For longitudinal transport of evaporated moisture, there is a distinct pattern of westward transport in the equatorial regions where the average travelled distance tends to exceed 10° with maxima up to 18°. Also for the backward analysis this pattern is clearly present and the maxima go up to 18°. Furthermore, there is eastward transport in the mid-latitudes, also with distances around 18°, caused by the trade winds. For the backward analysis, maximum values are smaller around the equator and in the mid-latitudes. The average travelled distance in
longitudinal direction is evidently larger than that in the latitudinal direction as atmospheric moisture fluxes in the east-west direction are typically stronger than in the north-south direction. The vertically integrated moisture flux obtained from ERA5 is indicated with arrows in Figures 6A,B and 7A,B. In general, the direction of the arrows coincides with the direction of the evaporated moisture transport obtained with UTrack. However, the magnitude of the arrows and the average travelled distance in longitudinal and latitudinal direction show some differences. For example, over northern Africa, the magnitude of the arrows
is small, but the average travelled distance in longitudinal direction is large. Because there is little evaporation in this region, the moisture flux is small; however, due to the strong trade winds the small amount of moisture is transported over a large distance, explaining the larger magnitudes for average transport in the longitudinal direction.

### 3.2 Validating checks

The sum of precipitation originating from any grid cell can be derived and used for its validation. For all source cells, the sum
of their downwind footprints (in fraction of evaporation) adds up to a value of around 1.05, due to the conversion from floats to integers. This means that the values should be renormalized and converted back to floats, so their global sum again equals one (see eq. 6). While this process reduces the data size considerably, it introduces some uncertainty in the footprints.

To further validate the dataset we compare the result of global precipitation (Figure 8A) with total precipitation from the ERA5 dataset. Because only ERA5 data are used to force the UTrack model, global precipitation should be similar between the two
datasets. Figure 8B shows the logarithm of the absolute difference between global precipitation derived with UTrack and from





ERA5. For most locations the error is small in comparison to the global precipitation. The largest deviations occur over mountainous regions and wet areas. Drier areas show a smaller absolute deviation, although the relative deviations can be large. Due to the relatively small error the precipitation patterns are similar for both datasets and both maxima and minima are located in the same regions. However, the exact values of extreme precipitation levels show some differences. For example,
precipitation at the coast of Colombia is higher for the ERA5 data.

## 4 Discussion

We present the first globally comprehensive dataset for atmospheric moisture flows at 0.5° (and 1.0°) spatial resolution. To produce the dataset we used an atmospheric moisture tracking model (UTrack) that simulates the three-dimensional atmospheric trajectories of a large amount of 'moisture parcels' from evaporation to precipitation (Tuinenburg and Staal,
2020). We forced our Lagrangian simulations with the latest ERA5 reanalysis data (Copernicus Climate Change Service, C3S). The ERA5 dataset is constrained by observations and represents the most detailed available representation of the atmosphere.

No independent observations exist to validate the source-to-sink relations in our database directly. Some studies validate moisture recycling using tracers in atmospheric models or stable water isotope measurements (Koster et al., 1993; Kurita and Yamada, 2008). However, tracers in atmospheric models are still very much dependent on model physics, especially the
treatment of the vertical transport of moisture (see Tuinenburg and Staal, 2020). Furthermore, because observations and interpretation of stable water isotopes are not available on a global scale, we do not use these here. However, this database provides the opportunity to analyze local stable water isotopic measurements in the light of the local moisture sources. Due to this inherent difficulty in validating the dataset, we provide an internal validating check regarding the global precipitation distribution. This showed that the patterns of total precipitation are similar to the total precipitation in ERA5, and are thus
internally consistent. However, due to the fact that the ERA5 climatological mean precipitation is about 1.5% higher than the evaporation, there is an added uncertainty of that order due to this non-closure of the moisture balance. The current dataset only stores the 2D downwind precipitation location and not the altitude at which the atmospheric moisture condensates. In future datasets, this could be stored and the resulting vertical latent heating profiles could be compared against the ERA5 latent heating profiles as a further validating check.

We analyzed a number of hydrologically relevant statistics of the global atmospheric moisture flows for 2008−2017. On average, 70% of global land evaporation rains down over land. This fluctuates across the year, with a minimum of 62% in December−January and a maximum of 74% in June−July. This evaporation recycling is higher than precipitation recycling: 51% of global land precipitation has evaporated from land, varying between 36% in January and 57% in June-July. This estimate is significantly higher than the estimate of 36% by Van der Ent et al. (2014), which was based on other input data and
a coarser, Eulerian, moisture tracking model. Highest basin recycling occurs in the Amazon and Congo basins, with evaporation and precipitation recycling of 63% and 36% for the Amazon basin and 60% and 47% for the Congo basin. These



numbers allow us to further compare our estimates of moisture recycling to previous ones. A number of studies have estimated the precipitation recycling ratio for the Amazon. These estimates tend to vary between 24−35% (Brubaker et al., 1993; Eltahir and Bras, 1994; Costa and Foley, 1999; Trenberth, 1999; Zemp et al., 2014; Staal et al., 2018), meaning that our new estimate is slightly above other ones, although a possible recycling ratio of 41% has also been reported (Burde et al., 2006). A recent estimate using a very similar Lagrangian moisture tracking method but forced with ERA-Interim data instead of ERA5 (Staal et al., 2018) estimated 32% evaporation recycling for the Amazon. This could mean that previous estimates based on the ERA-Interim dataset tend to underestimate atmospheric moisture recycling, although the study periods were not equal. Keune and Miralles (2019) found evaporation basin recycling values for the Danube of around 35% for JJA. For these months, we find a similar number, although the annual mean basin recycling rate is somewhat lower.

There are missing values for Antarctica and Greenland regarding precipitation and recycling ratios, which affect the metrics of travelled distance resulting in large gradients. These errors are the result of positive values in the ERA5 evaporation data, indicating condensation, which may represent errors in that dataset.

Recently, Link et al. (2020) developed another dataset for the destinations of global land evaporation. This current dataset differs in a number of important ways from that of Link et al. (2020). First, we present moisture connections among all global grid cells including the oceans, allowing the user to quantify the complete upwind evaporation source area across the globe. Second, our moisture tracking model (Tuinenburg and Staal, 2020) uses the most detailed forcing data that are currently available: hourly evaporation and precipitation, and hourly wind speed and three-dimensional wind directions for 25 atmospheric layers in the troposphere at 0.25° horizontal resolution from ERA5 (Copernicus Climate Change Service, C3S). This high-resolution model allowed us to generate the high-resolution output that we presented here. The moisture tracking model behind the Link et al. (2020) dataset (Van der Ent et al., 2014) uses ERA-Interim data with lower spatial and temporal resolution than ERA5.

The drawback of our high-resolution simulations is the heavy data load that results from them. This is the reason that we reduced the output resolution to 0.5°, as well as to 1.0° for further ease of handling the data. Users who wish to obtain moisture connections at 0.25° we refer to Tuinenburg and Staal (2020), who published the model code. Also, here we publish only (monthly) climatological means rather than time series. For time series from UTrack we again refer back to the model.

We foresee a range of applications of the dataset from fundamental Earth system science to applied global change research. For example, the moisture flows can be analyzed as a network where its nodes refer to areas that are connected by the moisture flows among them (Zemp et al., 2014; Krönke et al., 2020). Network analysis is becoming an increasingly important means to study global weather patterns and the Earth system (e.g. Boers et al., 2019). Our simulations potentially provide a mechanistic basis for empirical relations among remote weather patterns, such as drought correlations, and land cover. Finally, they can be

used to assess the effects of land cover changes on precipitation patterns at any scale between 0.5° (roughly 50 km at the equator) and the global scale.

## 5 Conclusions

Over the last decades, the interest in the atmospheric moisture cycle and the explicit links between evaporation and precipitation locations has increased. Moisture recycling rates are relevant for land-use change studies, hydrology, and atmospheric predictability. Many moisture recycling studies, including this study, determine these explicit relations between evaporation and precipitation location by forcing a moisture tracking model with atmospheric reanalysis data, effectively calculating moisture recycling as a post-processing scheme. Here, we have calculated the explicit relation between evaporation

and precipitation locations, based on the state-of-the-art, high-resolution ERA5 atmospheric reanalysis. The presented global database consists of the downwind precipitation location for all evaporation globally, on a 0.5° resolution for both the source locations and downwind precipitation location. It is presented as monthly mean values, averaged over 2008−2017. In addition to the data at 0.5°, a version at 1.0° resolution can be downloaded.

This dataset can be used to assess source-target relations of moisture fluxes into and out of the atmosphere. With these relations, the downwind precipitation and 2D latent heating effects of land-use changes could be determined. Furthermore, due to the global domain of the dataset, the same can be done for reverse direction, determining the evaporative moisture sources of precipitation in a certain location. These upwind moisture source locations could be used in atmospheric predictability studies.

## 6 Data availability

The dataset (Tuinenburg et al., 2020) is available from the PANGAEA archive in both the full resolution version at 0.5° resolution and in a lower resolution version at 1.0° resolution (https://doi.pangaea.de/10.1594/PANGAEA.912710). Sample scripts to open and process the database are provided on Github (https://github.com/ObbeTuinenburg/UTrack_global_database).

## Author contributions

OAT and AS conceived and designed the study. OAT and JJET carried out the study. All authors wrote the paper.

## Acknowledgments

OAT acknowledges support from the research program Innovational Research Incentives Scheme 016.veni.171.019 by the Netherlands Organisation for Scientific Research (NWO). AS acknowledges support from the European Research Council





project Earth Resilience in the Anthropocene (743080 ERA). Part of this work was performed in the cooperation framework
of Wetsus, European Centre of Excellence for Sustainable Water Technology (www.wetsus.eu). Wetsus is co-funded by the
Dutch Ministry of Economic Affairs and Climate Policy, the Northern Netherlands Provinces, the Province of Fryslan. The
authors would like to thank the participants of the Natural water production theme for the fruitful discussions and financial
support.

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



## Tables and figures

**Table 1: Average evaporation and precipitation basin recycling ratios for the 26 largest basins of the world.**

| River basin | Evaporation recycling ratio [-] | Precipitation recycling ratio [-] |
|---|---|---|
| Amazon | 0.63 | 0.36 |
| Amur | 0.39 | 0.32 |
| Chad | 0.24 | 0.21 |
| Congo | 0.60 | 0.47 |
| Danube | 0.25 | 0.17 |
| Ganges | 0.47 | 0.25 |
| Huang | 0.28 | 0.27 |
| Indus | 0.36 | 0.24 |
| Kolyma | 0.22 | 0.17 |
| Lawrence | 0.17 | 0.14 |
| Lena | 0.38 | 0.32 |
| Mackenzie | 0.33 | 0.34 |
| Mississippi | 0.31 | 0.25 |
| Murray | 0.11 | 0.12 |
| Nelson | 0.20 | 0.20 |
| Niger | 0.32 | 0.25 |
| Nile | 0.38 | 0.30 |
| Ob | 0.27 | 0.23 |
| Orange | 0.21 | 0.12 |
| Parana | 0.37 | 0.28 |
| Tigris | 0.17 | 0.06 |
| Volga | 0.19 | 0.16 |
| Yangtze | 0.44 | 0.30 |
| Yenisej | 0.33 | 0.26 |



| Yukon | 0.27 | 0.25 |
|---|---|---|
| Zambezi | 0.40 | 0.27 |

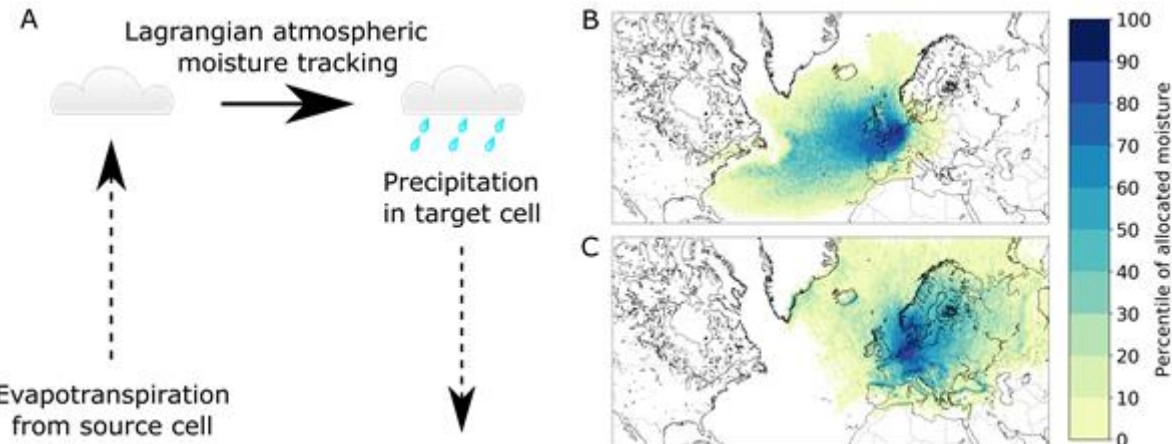


**Figure 1: The UTrack atmospheric moisture tracking model. A) The model tracks evaporation through the atmosphere from source cells to target cells using a Lagrangian moisture tracking scheme forced with ERA5 reanalysis data. B) An example of an 'evaporation footprint', or 'evaporationshed', from the UTrack model. C) An example of a 'precipitation footprint', or 'precipitationshed'. The examples show the distribution of evaporation that precipitated (B) and the precipitation that evaporated**

**(C) from Utrecht, the Netherlands, during 2008-2017, given as percentages of allocated moisture.**

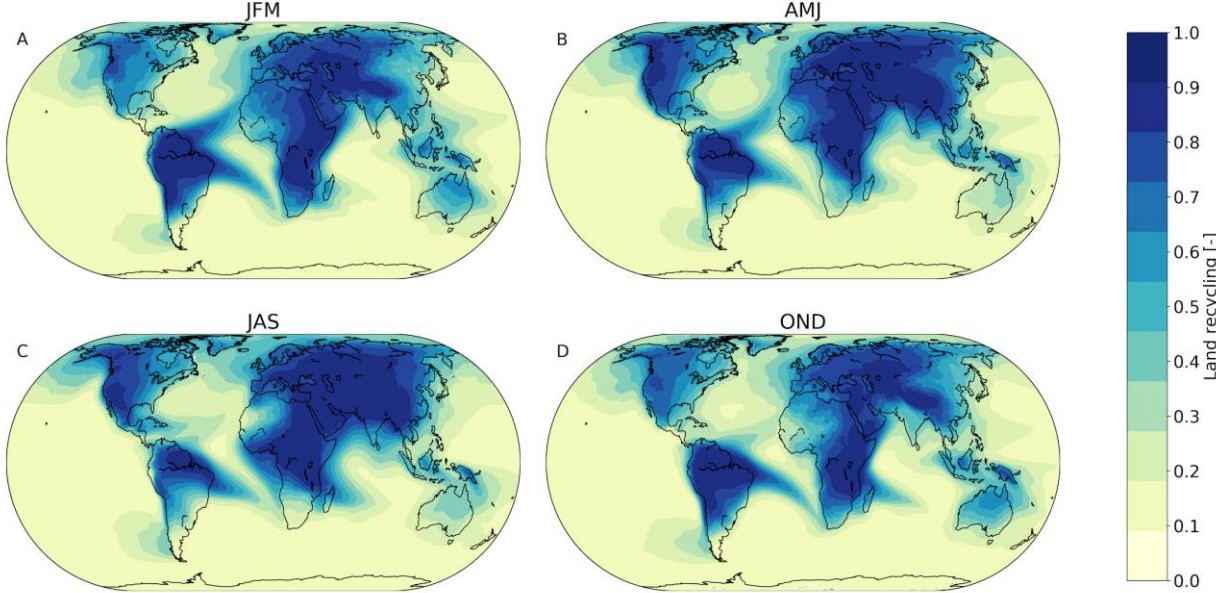

**Figure 2: Seasonal variability of evaporation recycling ratio (as a fraction) over land. Evaporation recycling ratio is the fraction of evaporation that subsequently rains down over land. A) January through March. B) April through June. C) July through September. D) October through December.**



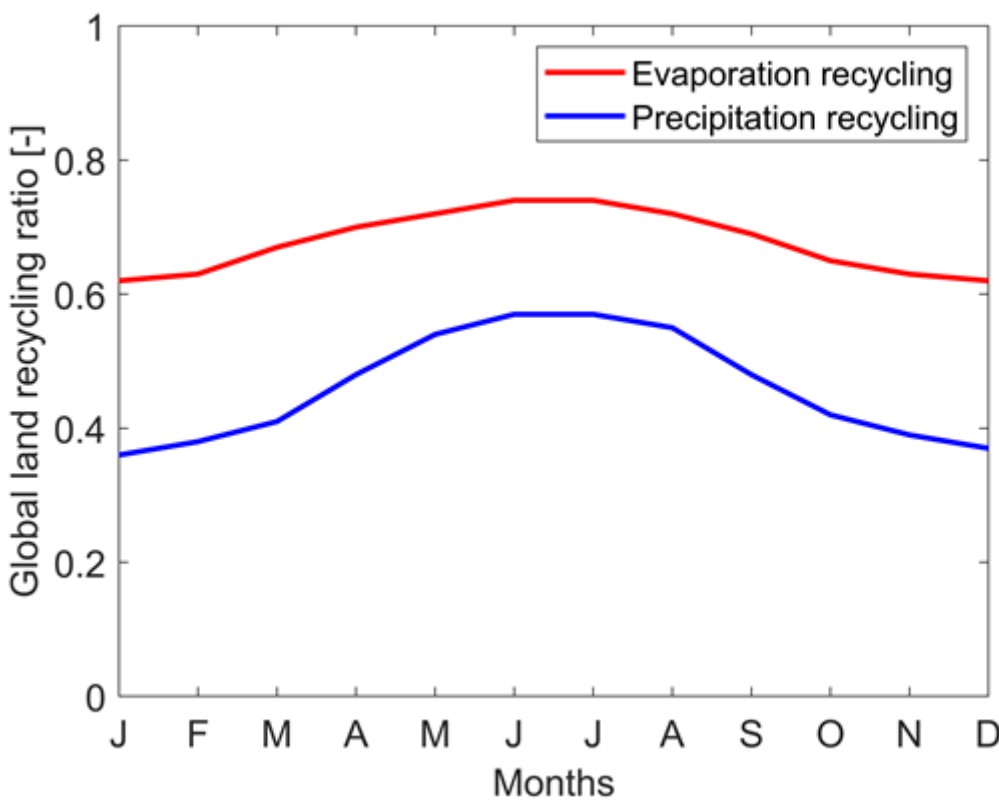

**Figure 3: Monthly variability of evaporation (red) and precipitation (blue) global land recycling ratios. Evaporation recycling ratio is the fraction of evaporation that subsequently rains down over land. Precipitation recycling ratio is the fraction of precipitation that has evaporated from land.**




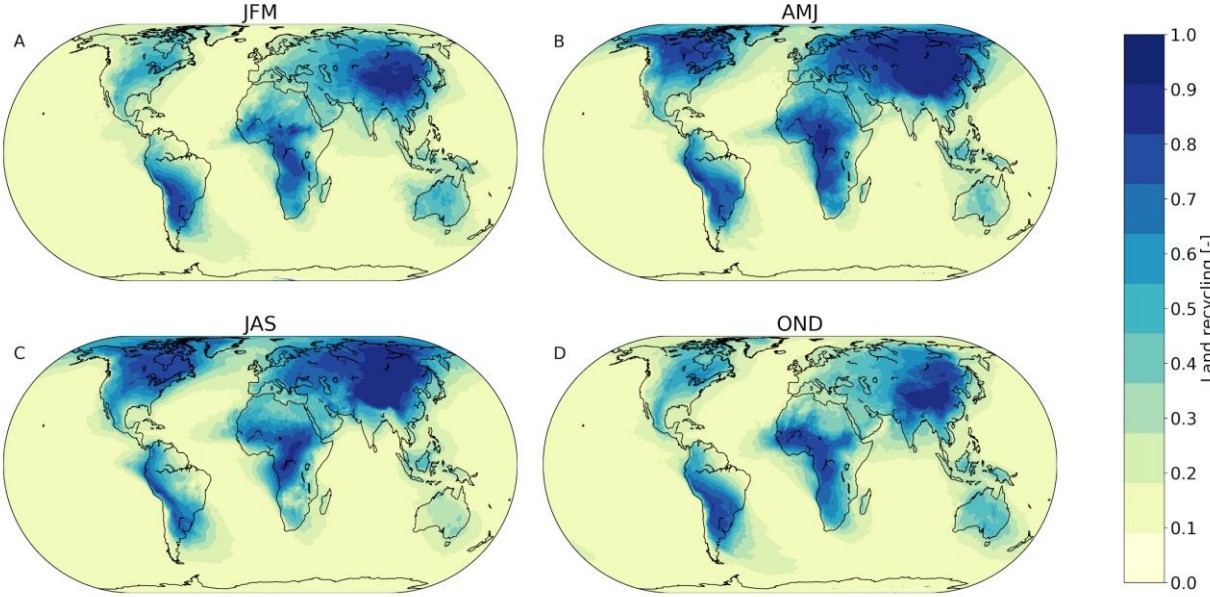

**Figure 4: Seasonal variability of precipitation recycling ratio over land. Precipitation recycling ratio is the fraction of precipitation that has evaporated from land. A) January through March. B) April through June. C) July through September. D) October through December.**




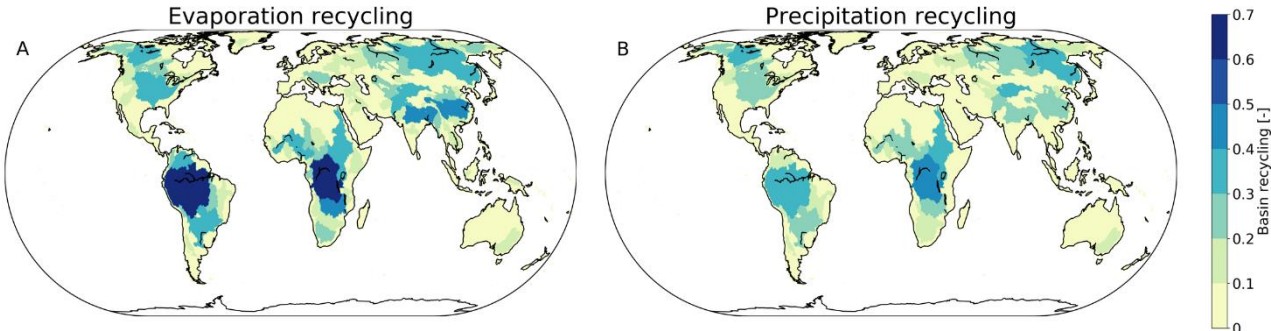

**Figure 5: Average evaporation (A) and precipitation (B) recycling ratios per basin. The basin recycling ratios are plotted for the 26 largest river basins.**


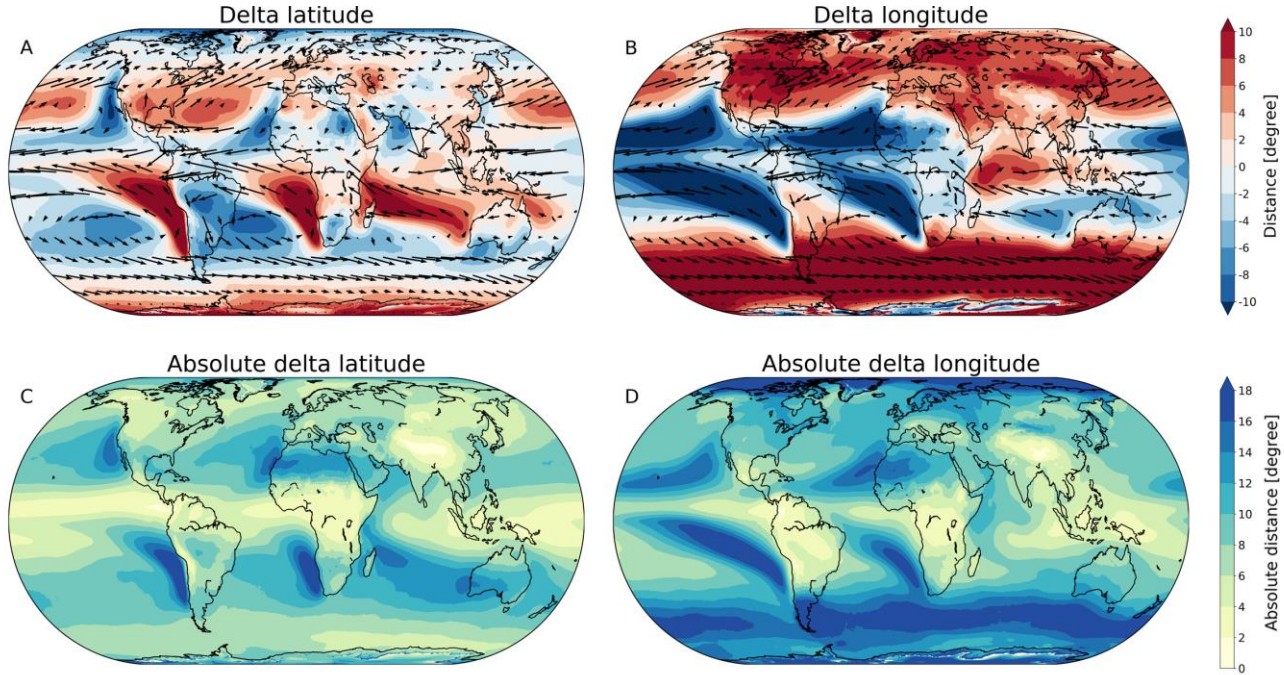

**Figure 6: Average forward moisture transport from evaporation to precipitation. A) Average of travelled distance in meridional direction ('delta latitude'). B) Average of travelled distance in zonal direction ('delta longitude'). C) Average of absolute travelled distance in meridional direction ('absolute delta latitude'). D) Average of absolute travelled distance in zonal direction ('absolute delta longitude').**

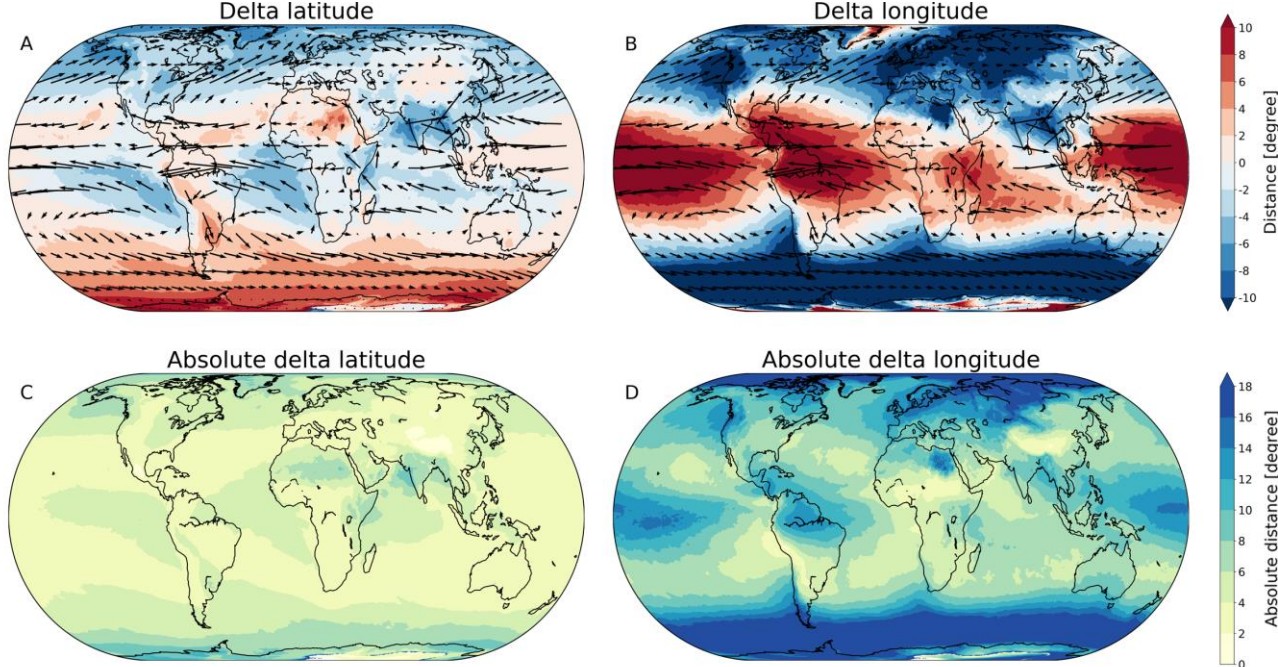

**Figure 7: Average backward moisture transport from precipitation to evaporation. A) Average of travelled distance in meridional direction ('delta latitude'). B) Average of travelled distance in zonal direction ('delta longitude'). C) Average of absolute travelled distance in meridional direction ('absolute delta latitude'). D) Average of absolute travelled distance in zonal direction ('absolute delta longitude').**

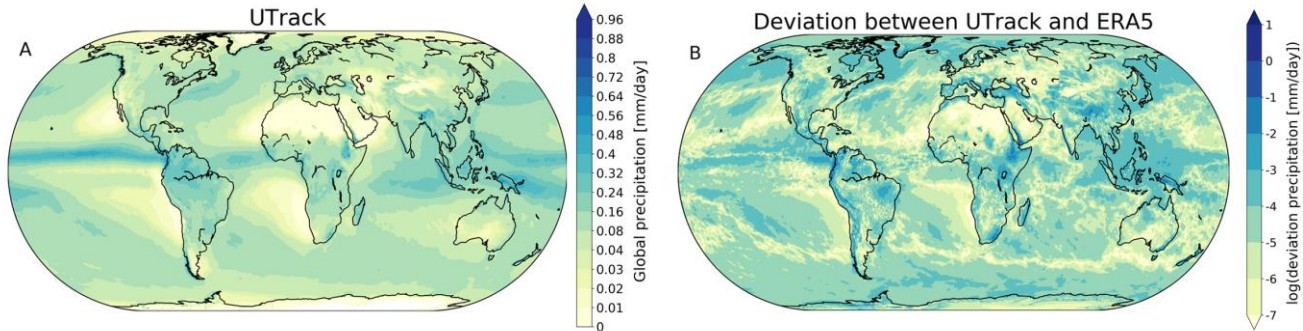

**Figure 8: A) Global precipitation derived with the moisture tracking model UTrack (this paper). B) Logarithm of the absolute deviation between global precipitation derived with UTrack and obtained from ERA5.**