# Peer review of "High-resolution global atmospheric moisture connections from evaporation to precipitation"

_Earth System Science Data, 2020_

## Referee Comment (RC1) · Anonymous Referee #1 · 15 Aug 2020

The study produced a globally complete dataset of atmospheric moisture flows from evaporation to precipitation based on ERA5 data. The paper is generally well-written and the data are useful. I have a few comments, mostly on the discussion of the results in the background of previous studies.

1. Please note the paper below. It also discussed nonlocal moisture contribution to precipitation. Therefore, the introduction around Line 35 and some other places should be careful.

Wei, J., & Dirmeyer, P. A. (2019). Sensitivity of Land Precipitation to Surface Evapo-transpiration: A Nonlocal Perspective Based on Water Vapor Transport. Geophysical

Research Letters, 46, 12,588–12,597. https://doi.org/10.1029/2019GL085613

The above paper also calculates the travelled distance of the moisture for precipitation but uses moisture content as weight (their Fig.3c,d). In this way, the very remote moisture, if in very tiny amounts, will have little effect on the average travelled distance. Is it more reasonable to use weights?

2. Section 2.1. It seems that you used a recently developed new moisture tracking method. In addition to the reference paper, can you summarize the advantages or differences of this method compared to other Lagrangian methods? According to your description, the method is similar to the QIBT back-trajectory method (Dirmeyer et al.) but is forward-trajectory.

3. About the evaporation recycling ratio and precipitation recycling ratio, I believe there are some previous studies. There should be some comparisons between your results and their results. To list a few:

Dirmeyer, P. A., J. Wei, M. G. Bosilovich, and D. M. Mocko, 2014: Comparing Evaporative Sources of Terrestrial Precipitation and Their Extremes in MERRA Using Relative Entropy, J. Hydrometeorology, 15, 102–116.

Van der Ent, R. J., Savenije, H. H. G., Schaefli, B. and Steele-Dunne, S. C.: Origin and fate of atmospheric moisture over continents, Water Resources Research, 46, W09525, doi:10.1029/2010WR009127, 2010.

4. Line 217-220. About the low recycling ratio in some basins, the explanation is not convincing. Actually, there have been studies on this. Generally, if the the remote moisture transfer is strong, such as in monsoon regions, the precipitation will be high and the recycling ratio will be low because the contribution from local evaporation is relatively small. For example, in Yangtze River basin, recycling ratio is higher (lower) in dry (wet) period. Refer to:

Wei, J., P. A. Dirmeyer, M. G. Bosilovich, and R. Wu, 2012: Water vapor sources

for the Yangtze River Valley rainfall: Climatology, variability, and implications for rainfall forecasting, Journal of Geophysical Research - Atmospheres, 117, D05126, doi: 10.1029/2011JD016902.

5. Line 137. Data stored in NetCDF4 format will be less precise? Or because you stored data into unsigned integers?

---

## Short Comment (SC1) · 21 Aug 2020

The authors did a great job in providing a global dataset with comprehensive information on the downwind areas of re-precipitation for evaporated water as well as for the upwind source areas of precipitation. Furthermore, it is the first global study of this sort using the recently published ERA 5 reanalysis data. The document is well written and understandable and provides interesting examples to exemplify possible uses.

I have a few comments as shown below:

1) The work does not provide a comparison of some of the results with previous work

[Figure]

and a rationale for occurring differences. The recycling numbers (global land evaporation which re-precipitates over land & global precipitation over land originating from land sources) seem to be comparatively high. While reading the manuscript, the reader might interpret that differences to previous studies might solely be due to the usage of better data (most actual reanalysis data and finer resolutions). However, it would perhaps also be relevant to relate those relatively high numbers to errors in the precipitation and evaporation. Figure 8b indicates within this context significant differences (Deviation between UTrack and ERA5).

2) The sample scripts seem not to work completely or there could be a bit more information on the necessary steps need to be done to get them running (e.g. how to derive the suitable net CDF file with monthly ERA5 data)

3) Minor comment to caption of Figure 1:

Perhaps the last phrase could be formulated a bit more precise in order to avoid misunderstandings:

For instance from:

"The examples show the distribution of evaporation that precipitated (B) and the precipitation that evaporated (C) from Utrecht, the Netherlands, during 2008-2017, given as percentages of allocated moisture."

To:

The examples show the distribution of re-precipitation for evaporated water from Utrecht (B) and the distribution of the city's sources of precipitation, given as percentages of allocated moisture.

4) Regarding the PANGAEA dataset, it would perhaps be a nice add-on to have the results also as yearly averages. But users might of course simply build them by their own and this should not be considered as a "must-have".

---

## Referee Comment (RC2) · Anonymous Referee #2 · 26 Aug 2020

Estimating the origin and the destination of the atmospheric moisture and its associated properties, such as travailing distance, is an interesting scientific question that also has a wide range of applications in water management, mitigations of climate change and weather forecasts. Via this study, authors offer a global monthly dataset on high resolution generated from a state-of-art tracking model and reanalysis dataset, ERA5. Analyses and data quality reported herein are convincing. I suggest an conception of this manuscript after the following comments being answered satisfactorily.

1. Line 88: Are these 25 model levels or pressure levels? 2. Lines 91-94: If authors run UTrack using global evaporation, then the distribution of atmospheric moisture in

each column should be similar to the Q (specific humidity) that obtained from ERA5 archive (if not on hourly scale, then it should be on daily and longer time scale). Is it the case? 3. Line 156: Conventionally, the precipitation recycling ratio is defined as \rho= P_et/P_tot. Therefore, what defined as the evaporation recycling ratio herein, is called the precipitation recycling ratio elsewhere, vice verse. I will suggest that either add a caveat to remind readers about this difference, or re-brand the term to follow the convention. 4. Lines 224-227: Can authors explains a bit more why the delta latitude is smaller in backward analysis? Why this is not shown in longitudinal transport? It is convincing as authors explain the difference in the local differences. How about this on global scale? 5. Both the precipitation and evaporation recycling ratios are high over the ragged topographies. Does this indicate that the mountains can intercept moisture flux transported from upstream and can trap evaporation originated locally? If this is the case, should we see a long/normal travailing distance in backward analysis from the upstream, and a short travailing distance in forward analysis to the downstream? I suggest a discussion on this point.
* * *

---

## Short Comment (SC2) · 2 Sep 2020

This paper presents a valuable contribution to the field of moisture recycling. Making the source code and the data openly available is very useful for other researchers interested in the application of these data. Being a co-author myself of a recently published similar dataset, also in ESSD (Link et al., 2020), I warmly welcome alternative datasets, which also give a sense to people outside the field what the general uncertainty is associated with different tracking methods and different data. The fact that the authors were able to provide the data on a relatively high 0.5° resolution highlights the efficiency of their code for calculations in case there is a single grid cell as source

region of interest.

That being said, I have one point of concern which has not been brought up by the reviewers so far. Figure 8 shows precipitation as calculated from Utrack (Fig. 8a) and the differences with ERA5 (Fig. 8b). First, I highly suspect either some typos or a calculation error in the postprocesing in Fig. 8a because global average precipitation should be in the range of 2-3 mm/day, which is thus 1 full order of magnitude higher than the scale in Fig. 8a. Second, I found the results in Fig. 8b difficult to interpret, because it is unclear what formula was exactly used to calculate the error and the log-aritmic scale is not intuitive. I suspect the formula to be $\log_{10}(P_{\mathrm{ERA}} - P_{\mathrm{Utrack}}$, which means that, for example, a difference of 0.1 mm/day becomes -1 and a difference of 1 mm/day is 0. A difference of 0.1 or 1 mm/day can be already a significant percentage of the total precipitation in some areas and this way of displaying the error does not allow to see whether there is a consistent under/overestimation of Utrack with respect to ERA5 or not. Moreover, it is unclear whether the differences are calculated on a daily/monthly/yearly/total time range scale, which would also influence the difference. In my opinion it would make sense if the authors would display two panels of relative and absolute error (with a diverging color scheme) and make it clear on which time scale these differences have been calculated. Besides, it would be useful if they would provide a global/land/ocean average error in order to identify whether there is a systematic bias or not, which might possibly explain why the recycling values found by the authors are relatively high compared to most previous research. Perhaps displaying the data in this way would simply confirm the authors' claim that the differences are small, but at least it will be easier to confirm for an independent reader.

Let me stress that it is very much appreciated that the authors bothered to calculate and show the errors in precipitation between ERA5 and Utrack in the first place, which is something I have not seen openly communicated in any other work with Lagrangian tracking schemes. I am convinced that the authors would be able to address my concern and I strongly support publication of this paper.

---

## Author Comment (AC1) · 30 Sep 2020

The study produced a globally complete dataset of atmospheric moisture flows from evaporation to precipitation based on ERA5 data. The paper is generally well-written and the data are useful. I have a few comments, mostly on the discussion of the results in the background of previous studies.

Thank you for the constructive feedback.

1. Please note the paper below. It also discussed nonlocal moisture contribution to precipitation. Therefore, the introduction around Line 35 and some other places should be careful.
Wei, J., & Dirmeyer, P. A. (2019). Sensitivity of Land Precipitation to Surface Evapotranspiration:
A Nonlocal Perspective Based on Water Vapor Transport. Geophysical Research Letters, 46, 12,588–12,597. https://doi.org/10.1029/2019GL085613
The above paper also calculates the travelled distance of the moisture for precipitation but uses moisture content as weight (their Fig.3c,d). In this way, the very remote moisture, if in very tiny amounts, will have little effect on the average travelled distance. Is it more reasonable to use weights?

Thank you for pointing us to this relevant paper. We now acknowledge this work in the introduction (line 30). We also rephrased lines 34-35 from "remain surprisingly poorly understood" to "are not fully understood".

We agree that the average travelled distance of moisture needs to be weighted by moisture content, and this is already the case in our calculations, as stated in line 163.

2. Section 2.1. It seems that you used a recently developed new moisture tracking method. In addition to the reference paper, can you summarize the advantages or differences of this method compared to other Lagrangian methods? According to your description, the method is similar to the QIBT back-trajectory method (Dirmeyer et al.) but is forward-trajectory.

The main difference with previous Lagrangian models is that UTrack uses the new ERA5 reanalysis data. The model settings are based on a sensitivity analysis given these forcing data. We made the novelty of the model more clear in Section 2.1 (line 75): "the first to be based on ERA5 atmospheric reanalysis data". For details on the model settings, we refer to Tuinenburg & Staal (2020).

Indeed, the simulations are based on forward tracking, but the model is also capable of backward tracking (see also lines 55-57).

3. About the evaporation recycling ratio and precipitation recycling ratio, I believe there are some previous studies. There should be some comparisons between your results and their results. To list a few:
Dirmeyer, P. A., J. Wei, M. G. Bosilovich, and D. M. Mocko, 2014: Comparing Evaporative Sources of Terrestrial Precipitation and Their Extremes in MERRA Using Relative Entropy, J. Hydrometeorology, 15, 102–116.
Van der Ent, R. J., Savenije, H. H. G., Schaefli, B. and Steele-Dunne, S. C.: Origin and fate of atmospheric moisture over continents, Water Resources Research, 46, W09525, doi:10.1029/2010WR009127, 2010.

Thank you for these suggestions. We compare our evaporation and precipitation recycling ratios with the literature in lines 282-299. We added reference to the ratios found by Dirmeyer et al. (2014) in lines 287-289: "Furthermore, Dirmeyer et al. (2014) found similar variability in the patterns and values of precipitation recycling throughout the year. However,

their results show some differences, including lower precipitation recycling in parts of South America." Rather than comparing our results with those from Van der Ent et al. (2010), we compare them with Van der Ent et al. (2014), which uses a more recent version of the WAM Eulerian tracking model.

4. Line 217-220. About the low recycling ratio in some basins, the explanation is not convincing. Actually, there have been studies on this. Generally, if the the remote moisture transfer is strong, such as in monsoon regions, the precipitation will be high and the recycling ratio will be low because the contribution from local evaporation is relatively small. For example, in Yangtze River basin, recycling ratio is higher (lower) in dry (wet) period. Refer to:
Wei, J., P. A. Dirmeyer, M. G. Bosilovich, and R. Wu, 2012: Water vapor sources for the Yangtze River Valley rainfall: Climatology, variability, and implications for rainfall forecasting, Journal of Geophysical Research - Atmospheres, 117, D05126, doi: 10.1029/2011JD016902.

Thank you for pointing this out. We added this as an additional explanation for the low recycling ratios in the mid-latitudes in lines 220-222: "Furthermore, differences in precipitation recycling can be expected due to regional and temporal differences in the strength of transport from moisture evaporated from the ocean, such as in monsoon regions (Wei et al., 2012)." Also, we now start line 217 with "In general, lowest recycling ratios are found…"

5. Line 137. Data stored in NetCDF4 format will be less precise? Or because you stored data into unsigned integers?

It is because we stored the data as unsigned integers that some imprecision was introduced (lines 139-140).

---

## Author Comment (AC2) · 30 Sep 2020

The authors did a great job in providing a global dataset with comprehensive information on the downwind areas of re-precipitation for evaporated water as well as for the upwind source areas of precipitation. Furthermore, it is the first global study of this sort using the recently published ERA 5 reanalysis data. The document is well written and understandable and provides interesting examples to exemplify possible uses.
I have a few comments as shown below:

Thank you for the encouraging words and constructive remarks.

1) The work does not provide a comparison of some of the results with previous work and a rationale for occurring differences. The recycling numbers (global land evaporation which re-precipitates over land & global precipitation over land originating from land sources) seem to be comparatively high. While reading the manuscript, the reader might interpret that differences to previous studies might solely be due to the usage of better data (most actual reanalysis data and finer resolutions). However, it would perhaps also be relevant to relate those relatively high numbers to errors in the precipitation and evaporation. Figure 8b indicates within this context significant differences (Deviation between UTrack and ERA5).

We agree that it is interesting to intercompare the recycling estimate, but the main goal of the current paper is the presentation of the dataset rather than this intercomparison. As the reviewer correctly points out, the literature on moisture recycling estimates uses different models and forcing datasets, and both these aspects contribute to the differences in moisture recycling estimates. Therefore, such a model intercomparison would require a more proper experiment, with different recycling models being forced with the same data. Nevertheless, we can speculate about the sources of the uncertainty in these intercomparisons. There are roughly two kinds of off-line moisture recycling models: those that focus on preserving the surface flux water balance (Eulerian schemes) and those that focus on preserving the moisture divergence balance as much as possible (Lagrangian schemes). In general, reducing the errors in one part of the atmospheric moisture path will increase these in the other part, but there may be a Pareto optimum. Unfortunately, the state of the art is that the moisture recycling studies do not report the errors in the moisture balance rigorously. (This also holds for our (OT and AS) previously published work.) By explicitly reporting the errors in our method (in Figure 8 in the current manuscript and by showing moisture tracking sensitivities in Tuinenburg and Staal, 2020 (HESS)), we hope to open the debate about such errors in the larger moisture tracking community.

2) The sample scripts seem not to work completely or there could be a bit more information on the necessary steps need to be done to get them running (e.g. how to derive the suitable netCDF file with monthly ERA5 data)

Thank you for this feedback. We added some more information in the sample scripts. Additionally, we added a script (ERA5_formatter.py) that can be used to sort the ERA5 data per month and to regrid the data to match the resolution of our dataset. These steps are also included in all other scripts that include ERA5 data.

3) Minor comment to caption of Figure 1:
Perhaps the last phrase could be formulated a bit more precise in order to avoid misunderstandings:
For instance from:
"The examples show the distribution of evaporation that precipitated (B) and the precipitation that evaporated (C) from Utrecht, the Netherlands, during 2008-2017, given as percentages of allocated moisture."
To:

The examples show the distribution of re-precipitation for evaporated water from Utrecht (B) and the distribution of the city's sources of precipitation, given as percentages of allocated moisture.

Thank you. We changed this phrase to: "The examples show the distribution of re-precipitation for evaporated water from Utrecht (the Netherlands) during 2008−2017 (B) and the distribution of the city's source of precipitation (C), given as percentages of allocated moisture" (lines 441-442).

4) Regarding the PANGAEA dataset, it would perhaps be a nice add-on to have the results also as yearly averages. But users might of course simply build them by their own and this should not be considered as a "must-have".

We provide a script called yearly_average.py which calculates that. In this way we minimize the total size of the dataset(s).

---

## Author Comment (AC3) · 30 Sep 2020

Estimating the origin and the destination of the atmospheric moisture and its associated properties, such as travailing distance, is an interesting scientific question that also has a wide range of applications in water management, mitigations of climate change and weather forecasts. Via this study, authors offer a global monthly dataset on high resolution generated from a state-of-art tracking model and reanalysis dataset, ERA5. Analyses and data quality reported herein are convincing. I suggest an conception of this manuscript after the following comments being answered satisfactorily.

Thank you for these positive words.

1. Line 88: Are these 25 model levels or pressure levels?

They are pressure levels, which we now clarified in line 88. Note that the model itself, being Lagrangian rather than Eulerian, does not contain individual levels or grid cells.

2. Lines 91-94: If authors run UTrack using global evaporation, then the distribution of atmospheric moisture in each column should be similar to the Q (specific humidity) that obtained from ERA5 archive (if not on hourly scale, then it should be on daily and longer time scale). Is it the case?

That is correct. As the model is forced by these data, and the moisture parcels are distributed vertically with the local moisture profile, they are indeed the same.

3. Line 156: Conventionally, the precipitation recycling ratio is defined as nrho= P_et/P_tot. Therefore, what defined as the evaporation recycling ratio herein, is called the precipitation recycling ratio elsewhere, vice verse. I will suggest that either add a caveat to remind readers about this difference, or re-brand the term to follow the convention.

We agree with this definition of precipitation recycling ratio, but please note that our evaporation recycling ratio refers to the fraction of evaporation that precipitates over land and is therefore different from mentioned definition of precipitation recycling. Hence, our definitions are consistent with those in the literature. To avoid confusion, we added the definitions for land and basin evaporation recycling as formulas in lines 157-158: "(epsilon_xy = P_E,xy,land / ET_xy)" and "(epsilon_xy = P_E,xy,basin / ET_xy)".

4. Lines 224-227: Can authors explains a bit more why the delta latitude is smaller in backward analysis? Why this is not shown in longitudinal transport? It is convincing as authors explain the difference in the local differences. How about this on global scale?

This has to do with the variability in the atmospheric moisture cycle intensity. Consider, for example, an oceanic site in a low precipitation area. In this location the evaporation will always be high and precipitation will always be low. Therefore, evaporation entering the atmosphere at that location will typically be transported far away, because the chance of precipitation is low. However, the small amount of precipitation that will fall in such a location will quite likely originate from evaporation close by, because there is so much evaporation in that location. The reverse reasoning will hold for a very wet location where P>>E. Any evaporation from that location will not be transported far away, but the precipitation there will come from far, as there is not much local evaporation. As mean precipitation has an upper bound that is much higher than the mean evaporation, there are many more regions where E>P than where P>E. Therefore, it appears on the maps in Figures 6 and 7 that there is a strong difference in distance between forward and backward tracking.

Note that this difference will not be present if the atmospheric moisture flow and E and P fluxes are relatively constant. So, in homogeneous atmospheric vertically integrated moisture transport and constant E=P, the upwind length scale is equal to the downwind length scale. We hypothesize that there is much more variability in the north-south direction than in the east-west direction, due to homogeneous east-west moisture flows in the tropics and mid-latitude, with erratic interaction between these two systems in the form of atmospheric river flows. This would explain the stronger difference between upwind and downwind length scale in the latitudinal direction than in the longitudinal direction. However, the details of these exact interactions could be explored in future studies using the presented dataset.
We have added this to the explanation in section 3.1 (lines 229-238):

"The difference between the forward and backward tracked distance can be explained by the variability in the atmospheric moisture cycle in several dimensions. In dry areas, the downwind length scale will be long, because any evaporation will travel very far before it rains out. The upwind length scale will be short because of the large amount of local evaporation relative to the local precipitation. Contrastingly, in wetter areas, the reverse is true and the upwind length scale will be longer: the precipitation falling in wet areas will have come from far away, because of the limited local evaporation relative to the precipitation. In these areas, the downwind length scale is shorter because evaporation from wetter areas will have a larger chance to be part of a nearby precipitation event. As can be seen in Figures 6 and 7, the difference between forward and backward distance typically is larger in the meridional than in the zonal direction. We hypothesize that this is due to the fact that there is less variability in the moisture transport in the zonal direction than in the meridional direction."

5. Both the precipitation and evaporation recycling ratios are high over the ragged topographies. Does this indicate that the mountains can intercept moisture flux transported from upstream and can trap evaporation originated locally? If this is the case, should we see a long/normal travailing distance in backward analysis from the upstream, and a short travailing distance in forward analysis to the downstream? I suggest a discussion on this point.

That is correct, as is illustrated by some differences between Figures 6 and 7. Take, for example, the "forward delta longitude" for the Andes (Fig. 6B), which is much lower than the "backward delta longitude (Fig. 7B). We added a sentence about this in the discussion (lines 320-322): "There could be a benefit in classifying the weather patterns in terms of the length scale difference between the upwind and downwind parts of the atmospheric moisture cycle."

---

## Author Comment (AC4) · 30 Sep 2020

This paper presents a valuable contribution to the field of moisture recycling. Making the source code and the data openly available is very useful for other researchers interested in the application of these data. Being a co-author myself of a recently published similar dataset, also in ESSD (Link et al., 2020), I warmly welcome alternative datasets, which also give a sense to people outside the field what the general uncertainty is associated with different tracking methods and different data. The fact that the authors were able to provide the data on a relatively high 0.5° resolution highlights the efficiency of their code for calculations in case there is a single grid cell as source region of interest.

Thank you for these encouraging words.

That being said, I have one point of concern which has not been brought up by the reviewers so far. Figure 8 shows precipitation as calculated from Utrack (Fig. 8a) and the differences with ERA5 (Fig. 8b). First, I highly suspect either some typos or a calculation error in the postprocesing in Fig. 8a because global average precipitation should be in the range of 2-3 mm/day, which is thus 1 full order of magnitude higher than the scale in Fig. 8a. Second, I found the results in Fig. 8b difficult to interpret, because it is unclear what formula was exactly used to calculate the error and the logaritmic scale is not intuitive. I suspect the formula to be log10(PERA $\square$ PUtrack, which means that, for example, a difference of 0.1 mm/day becomes -1 and a difference of 1 mm/day is 0. A difference of 0.1 or 1 mm/day can be already a significant percentage of the total precipitation in some areas and this way of displaying the error does not allow to see whether there is a consistent under/overestimation of Utrack with respect to ERA5 or not. Moreover, it is unclear whether the differences are calculated on a daily/monthly/yearly/total time range scale, which would also influence the difference. In my opinion it would make sense if the authors would display two panels of relative and absolute error (with a diverging color scheme) and make it clear on which time scale these differences have been calculated. Besides, it would be useful if they would provide a global/land/ocean average error in order to identify whether there is a systematic bias or not, which might possibly explain why the recycling values found by the authors are relatively high compared to most previous research. Perhaps displaying the data in this way would simply confirm the authors' claim that the differences are small, but at least it will be easier to confirm for an independent reader.

We are grateful for these attentive remarks about Figure 8. It turns out that we made an error in producing this figure. We multiplied the results that are presented in the article with a factor that is incorrect. (It should be noted that the total precipitation obtained from ERA5 was also multiplied with this incorrect factor before calculating the deviation.) The updated result shows an average global precipitation of 2.43 mm/day which agrees with the range indicated by Van der Ent. We replaced Figure 8 with the correct one. We also expanded this figure with two additional panels. It now includes the global precipitation from UTrack (in mm/day), that from ERA5, and their absolute and relative differences.

Let me stress that it is very much appreciated that the authors bothered to calculate and show the errors in precipitation between ERA5 and Utrack in the first place, which is something I have not seen openly communicated in any other work with Lagrangian tracking schemes. I am convinced that the authors would be able to address my concern and I strongly support publication of this paper.

Thanks again for helping to improve our paper.